# Metabolic Adaptation and Its Determinants in Adolescents Two Years After Sleeve Gastrectomy

**DOI:** 10.3390/nu17010075

**Published:** 2024-12-28

**Authors:** Vibha Singhal, Clarissa C. Pedreira, Shubhangi Tuli, Lea Abou Haidar, Ana Lopez Lopez, Meghan Lauze, Hang Lee, Miriam A. Bredella, Madhusmita Misra

**Affiliations:** 1Division of Pediatric Endocrinology, Massachusetts General Hospital, Harvard Medical School, Boston, MA 02114, USA; vibhasinghal@mednet.ucla.edu (V.S.); clarissaccarvalho@hotmail.com (C.C.P.); 2Neuroendocrine Unit, Massachusetts General Hospital, Harvard Medical School, Boston, MA 02114, USA; shubhangi.tuli@downstate.edu (S.T.); leaabouhaidar13@gmail.com (L.A.H.); alopezlopez@mgh.harvard.edu (A.L.L.); mlauze@mgh.harvard.edu (M.L.); 3Department of Pediatrics, Mattel Childrens’ Hospital, UCLA, Los Angeles, CA 90095, USA; 4Department of Pediatrics, Harvard Medical School, Boston, MA 02114, USA; 5MGH Biostatistics Center, Harvard Medical School, Boston, MA 02114, USA; hlee5@mgh.harvard.edu; 6Department of Radiology, Massachusetts General Hospital, Harvard Medical School, Boston, MA 02114, USA; miriam.bredella@nyulangone.org; 7Department of Radiology, NYU Langone Health, New York, NY 10016, USA; 8Department of Pediatrics, University of Virginia, Charlottesville, VA 22903, USA

**Keywords:** resting energy expenditure, adolescents, sleeve gastrectomy, metabolic adaptation, fat-free mass

## Abstract

**Background/Objective:** Weight loss is associated with reductions in resting energy expenditure (REE), which are impacted by changes in body composition following sleeve gastrectomy (SG). Current data regarding changes in measured REE (mREE) and metabolic adaptation in adolescents after SG are limited. We evaluated changes in mREE, metabolic adaptation, and body composition in youths after SG vs. non-surgical (NS) controls over two years. **Methods:** Youths 14–22 years old undergoing SG (*n* = 24) and NS controls with severe obesity (*n* = 28) were recruited. mREE was determined using indirect calorimetry. Predicted REE (pREE) was calculated using regression equation derived from baseline data of our cohort and used to calculate pREE at follow up. Metabolic adaptation was calculated as mREE − pREE. We normalized REE to fat-free mass (FFM) and total body weight (TBW). Dual energy X-ray absorptiometry was used to measure body composition. Measurements were performed at baseline and two-years. **Results:** Baseline age, sex, and BMI were similar between groups. Greater decreases in BMI in SG vs. NS (−12.4 (−14.4, −9.8) vs. 2.2 (0.3, 3.5) kg/m^2^, *p* < 0.0001) and within-group decreases in mREE (401.0 ± 69.5 kcal/d; *p* < 0.0001) in SG were seen. mREE/FFM decreased within the SG group (*p* = 0.006), the two-year change in mREE/FFM and mREE/TBW did not differ between groups (*p* = 0.14 and 0.24). There was no metabolic adaptation within SG. **Conclusions:** Despite significant decreases in BMI after SG in youths, no metabolic adaptation was present at two years. This implies that by two years, metabolism has reached a steady state and weight changes after this should be addressed in an unbiased way.

## 1. Introduction

The high prevalence and increase in severe obesity among children and adolescents is concerning and leads to increased health complications in adulthood [1,2]. Adolescents with severe obesity often have a long history of inability to attain and/or sustain weight loss with traditional weight-loss methods. This has been attributed to endocrine adaptations that might increase appetite and decrease satiety [3,4], as well as metabolic adaptations that support weight regain such as through a decrease in resting energy expenditure (REE), especially following caloric restriction. Metabolic and bariatric surgery (MBS), particularly sleeve gastrectomy (SG), is increasingly recognized as a safe and effective treatment for severe obesity in adolescents [5,6]. Weight loss magnitude and maintenance following surgical procedures surpass those of conventional methods [7]. However, some weight regain is common even after MBS following attainment of nadir weight [8], and 10–20% of patients struggle with considerable weight regain (75–90%) after MBS [9]. The reasons for the observed variability in weight loss (and improved metabolic outcomes) and subsequent weight regain among patients following SG remains unclear [10].

Resting energy expenditure (REE) represents basic energy requirements of daily life, which comprise approximately two thirds of total energy expenditure (TEE) [11]. REE is determined by age, sex, body size and composition, hormones, and the sympathetic nervous system. An excessive decline in REE is a major determinant of weight regain when weight loss is achieved by caloric restriction [12,13]. The greater decline in REE than expected for weight loss is likely consequent to compensatory metabolic processes that then prevent continued weight loss [14]. Disproportionate reductions in REE following reductions in fat-free mass (FFM), seen after MBS in some patients, may predispose them to weight regain [7] and thus impact long-term maintenance of weight loss after MBS. There are only a few studies that have evaluated changes in REE after SG, and results are conflicting likely from different methodologies used for evaluating REE, different patient populations (with or without diabetes), and different follow up periods [15,16]. Some studies in adults have found an increase in REE relative to total body mass postoperatively [17,18,19], whereas others have found reductions in REE during different postoperative periods [20,21]. Data regarding metabolic adaptation, and particularly REE, following MBS in adolescents and young adults are lacking.

The objectives of this study were as follows: (1) evaluate changes in measured REE (mREE) and body composition over two years, as well as metabolic adaptation at two years in youth after SG compared to non-surgical (NS) controls; and (2) examine associations of changes in REE and metabolic adaptation with changes in weight and body composition. We have expanded the cohort from our previously published study that examined one-year changes in REE [22] (and now report data for two-year changes in REE and metabolic adaptation in a larger cohort).

## 2. Participants and Methods

### 2.1. Participants

Between 2015 and 2020, 52 adolescents and young adults ranging from 14 to 22 years of age with moderate to severe obesity [23] were recruited from different obesity treatment centers in Massachusetts (primarily the Massachusetts General Hospital Weight Center, Brigham, and Women’s Hospital and Boston Medical Center). Moderate obesity was defined as BMI 120 to <140% of the 95th percentile for age and sex in those ≤ 18 years of age and 35 to <40 kg/m^2^ for those >18 years old. Severe obesity was defined as BMI ≥ 140% of the 95th percentile for age and sex in those ≤ 18 years of age and ≥ 40 kg/m^2^ for those >18 years old. Of note, all participants included had attained final adult height at the start of the study.

In total, 24 participants underwent SG and 28 were followed without surgery and routine lifestyle interventions (non-surgical (NS) participants). This study was approved by the Partners Healthcare Institutional Review Board and is Health Insurance Portability and Accountability Act (HIPAA) compliant. Prior to inclusion in the study, informed consent was obtained from participants 18 years and older and from parents of participants younger than 18 years. Informed assent was obtained from minors.

After a screening visit to confirm eligibility for the study, all participants underwent a complete medical history and anthropometric evaluation at baseline (BL) and at one and two years. Study participation did not interfere with routine clinical care of participants. Exclusion criteria for the study included the following: (a) use of antipsychotic medications that cause weight gain if treated for less than six months, or if the dosage was not stable for at least two months (ensuring they were on a stable dose and stable weight); (b) untreated thyroid dysfunction; (c) smoking > 10 cigarettes/day; (d) substance use disorder; and (e) pregnancy or lactation.

We present data on medications used by the participants at baseline and follow up. (Table 1). Of note, none of our participants were on glucagon-like-peptide 1 receptor agonists (GLP1R agonists) before or during the study, as this study was performed before these became available for use in adolescents.

### 2.2. Clinical and Anthropometric Variables

Subjects were asked to fast for at least eight hours prior to each visit and were asked to avoid heavy exercise, alcohol, or tobacco use the day before the visit. They were asked to continue their usual dietary habits. Weight (kg), height (m), and body mass index (BMI) (kg/m^2^) were determined. Height was measured to the nearest 0.1 cm using a wall-mounted stadiometer as the mean of three measurements, and weight was measured to the nearest 0.1 kg using an electronic scale. BMI was calculated as weight (kg) divided by the height-squared (m^2^). BMI z-score and BMI % of the 95th percentile for age and sex were calculated using the NCHS 2000 standards [24]. Percent weight loss was calculated for the different timepoint changes (one year–BL, two years–BL) using the formula [(weight at timepoint 2 − weight at timepoint 1)/weight at timepoint 1] × 100). Activity levels and sleeping hours were assessed using the Paffenberger questionnaire [25]. Paffenberger questionnaire is a self-reported questionnaire that assesses vigorous, moderate, and light physical activity and sleep duration. It has good validity and excellent reliability in measuring physical activity when compared to accelerometers.

### 2.3. Body Composition

Fat mass (FM) and FFM were assessed using dual-energy X-ray absorptiometry (DXA) (Hologic QDR 4500, Hologic Inc., Waltham, MA, USA) at each timepoint. The coefficients of variation are 1.1% for FM and 2.1% for FFM, respectively, at our institution. Percentage of FM and FFM were calculated as (FM and FFM divided by total body mass) × 100. The fat-free mass index (FFMI) was calculated as FFM (kg) divided by height-squared (m^2^).

### 2.4. Energy Expenditure

Indirect calorimetry was used to measure fasting mREE and respiratory quotient (RQ) at each timepoint. It was performed under thermal neutrality using the VMAX Encore 29 metabolic cart (Viasys Healthcare, Carefusion; San Diego, CA, USA). All measurements were performed at the Translational and Clinical Research Center of our institution. As described by Wolfe et al., to assess predicted REE (p REE), we used the baseline data of our cohort in a linear regression model using mREE as the dependent variable and FFM as the independent variable. We used the equation generated to obtain a predicted REE using FFM for each participant at both follow-up timepoints [26].

The specific equation generated was as follows: 171.23 + 24.02 (total fat-free mass in Kg).

The difference between mREE and pREE was calculated to estimate metabolic adaptation. We calculated mREE divided by the fat-free mass (mREE/FFM) in kcal/d/kg and mREE divided by the total body weight (TBW) (mREE/TBW) in kcal/d/kg for each participant to control for the effect of FFM or TBW on REE.

### 2.5. Statistical Analysis

Data were analyzed using JMP Statistical Discovery Software (JMP Pro Version 16). Results are reported as mean ± standard error of mean (SEM) for normally distributed data or median (first quartile, third quartile) for data not normally distributed. For continuous variables, depending on data distribution, we applied either Student’s *t*-test (for parametric data) or a 2-sample Wilcoxon rank sum test (for non-parametric data) to compare between-group differences and a paired samples *t*-test or Wilcoxon signed rank test to compare within-group changes over time. To assess for within-group metabolic adaptation, we used the one sample *t*-test to determine whether the mean was different from zero [26]. Fishers Exact test was used to compare proportions. A two-tailed *p*-value < 0.05 was considered significant. Univariate Spearman correlational analyses were used to determine associations of the two-year change in REE and metabolic adaptation at two years with two-year change in weight and body composition over this duration. A *p*-value of <0.05 was used as the threshold for statistical significance.

## 3. Results

### 3.1. Subject Characteristics

A total of 24 adolescents electing for SG and 28 NS controls participated in the study. Baseline demographic and anthropometric characteristics are shown in Table 2. SG vs. NS participants did not differ in age, sex, race, ethnicity, height, and weight. Median BMI and BMI, expressed as percentage of the 95th percentile of BMI for age, was nominally higher (without reaching statistical significance) in the SG group compared to the NS group; however, BMI z-score did not differ between groups at baseline. Groups did not differ for number of hours of sleep and for exercise activity.

### 3.2. Baseline Body Composition and REE

Baseline measurements of body composition (by DXA) and REE in the SG and NS groups are shown in Table 3. Percent body fat was higher and percent lean mass was lower in the SG group compared to NS. mREE did not differ between the two groups at baseline.

### 3.3. Changes in Anthropometric and Body Composition Measures over Two Years

Anthropometric and body composition changes over two years are shown in Table 4. All anthropometric measures and body composition decreases were higher in the SG than the NS group from BL to one year and BL to two years. Changes in BMI and fat-free mass over two years in the SG and NS groups are shown in Figure 1A and Figure 1B, respectively. Within-group reductions in BMI and fat-free mass following sleeve gastrectomy were most marked one year following surgery, after which these measures stabilized between one and two years. In contrast, the non-surgical group had within group increases in BMI and fat-free mass over two years.

### 3.4. REE and Metabolic Adaptation over Two Years

Changes in mREE and mREE standardized for FFM over two years are shown in Figure 2A and Figure 2B, respectively. Within-group reductions in mREE following sleeve gastrectomy were most marked one year following surgery in the SG group, after which this measure stabilized (between one and two years). The NS group demonstrated no change in mREE over one and two years. The SG and NS groups differed significantly for mREE at one and two years.

mREE standardized for FFM decreased within the SG group over one and two years, but this measure did not significantly differ from the NS group at the specific timepoints. mREE standardized for TBW did not change in the surgical or non-surgical group over one or two years. Changes in mREE over two years remained significant after controlling for participants who were on ADHD and weight loss medications at any time over the course of this study.

Metabolic adaptation (estimated as a difference between measured and predicted REE) was lower than zero in the SG group at one year (−106.94 ± 31.31; *p* = 0.003), but not at two years (−86.71 ± 65.92; *p* = 0.204). In the NS group, there was no metabolic adaptation seen at one year (−21.72 ± 29.29; *p* = 0.47) or at two years (4.6 ± 36.67; *p* = 0.90), i.e., mean was not statistically different from zero. SG and NS groups had a statistical trend to show differences for metabolic adaptation at one year (*p* = 0.053), but not at two years (*p* = 0.207).

### 3.5. Correlation of Changes in REE, Weight Loss, Body Composition, and Metabolic Adaptation at Two Years

Within SG, metabolic adaptation at two years was positively associated with two year changes in mREE/FFM (Spearman ρ = 0.47 *p* = 0.04). Within the NS group, metabolic adaptation at two years was positively associated with two year changes in mREE (Spearman ρ = 0.54 *p* = 0.004) and with two year changes in mREE/FFM (Spearman ρ = 0.46 *p* = 0.02). There was no correlation of changes in weight, BMI, fat mass, and fat-free mass with metabolic adaptation at two years.

## 4. Discussion

We show that REE decreases over two years following SG in adolescents with obesity, and that within the SG group, there is metabolic adaptation at one year which is no longer prevalent two years after surgery. Furthermore, metabolic adaptation is positively associated with two-year changes in measured REE (mREE) that are normalized for changes in fat-free mass. However, the two-year changes in measured REE (normalized for fat-free mass and total body weight) and metabolic adaptation did not differ between surgical and non-surgical groups.

There are very few studies of energy expenditure in adolescents following bariatric surgery with a 2-year follow up. One longitudinal controlled trial in 11 adolescents with obesity reported that basal metabolic rate declined by 1.5 months and remained suppressed at 6 and 12 months after Roux-en-Y gastric bypass (RYGB) [27]. Metabolic adaptation was not investigated in that study. Another longitudinal study with 12 adolescents and young adults used indirect calorimetry to show that REE decreased over the first year following surgery, consistent with a decrease in total lean mass; however, REE standardized for total body weight increased one year after SG [22]. In another study of 20 adolescents undergoing RYGB or SG, mREE decreased by 548 kcal/d one year after surgery [28]. We found similar decreases in measured REE in our study (decreased by 401 kcal/d two years) after SG. Furthermore, we found that there was metabolic adaptation (measured REE was lower than predicted REE based on body composition) at one-year, but this was no longer prevalent at two years. Metabolic adaptation was present during the first year after surgery during the active weight loss stage and then stabilized at two years when weight loss plateaued, despite a lower weight state compared to presurgical weight.

Significant reductions in REE normalized for fat-free mass and metabolic adaptation after MBS could be one of the contributing factors to the variable weight loss seen after sleeve gastrectomy and may contribute to weight regain [28]. According to one study, it is possible that changes in muscle mass that occur after MBS could play a crucial role in alterations in REE, as well as the degree of weight loss and weight loss maintenance [29]. Data suggest that, irrespective of the modality of weight loss—very-low-calorie diet or bariatric surgery—the degree of metabolic adaptation is associated with the degree of increase in appetite [30]. This is significant clinically, as changes in appetite after MBS can be monitored and may be predictive of weight loss.

Weight loss during the first year after surgery includes reductions in fat mass and fat-free mass, with greater reductions occurring in fat mass as compared to fat-free mass. Most of the reductions in the metabolically active fat-free mass occur from reductions in skeletal muscle [31]. Some studies have found that the REE/FFM ratio decreases after MBS [11], while the REE/TBW ratio increases [11,22], postulating that the decrease after surgery in REE/FFM could be due to a decline in FFM metabolism, while the increase in REE/TBW following surgery is primarily driven by the substantial loss of FM [11]. We report similar results in our study, with decreases in mREE/FFM but no changes in mREE/TBW found at one and two years after SG.

Data on metabolic adaptation after MBS in adults are conflicting, with studies reporting decreases [15,29], increases [32,33,34], or no changes [10,35,36] in metabolic adaptation post-surgery. One study reported that metabolic adaptation at six months was different from baseline, with no further changes between six months and two years [27]. Consistent with the literature, in our study, we report metabolic adaptation at one year, but not at two years after sleeve gastrectomy. The inconsistency in results across studies likely originates from the duration of follow-up, with shorter follow-up studies reporting metabolic adaptation when there is a physiological effort to reduce/prevent weight loss, and this is not seen in longer-term (one year and longer) studies. Differences in methodologies to estimate REE and metabolic adaptation [37], small sample size [38], and differences in the weight loss modality are other possible explanations for conflicting data.

In our study, we show that the degree of metabolic adaptation is associated with changes in REE normalized to fat-free mass. Hence, decreasing the loss of fat-free mass and making the skeletal muscle more metabolically active after surgery may help decrease metabolic adaptation and improve weight loss after surgery. Engagement in physical exercise is associated with greater weight loss after MBS [39,40]. Activity levels and sleeping hours could be factors influencing postoperative changes in REE. However, in our study, neither vigorous exercise nor sleeping hours differed between groups at baseline or follow-up. A high-protein diet is recommended after surgery and is considered to be helpful in preserving lean mass. However, a recent study did not show the preservation of lean mass after protein supplementation in post-bariatric patients [41].

The strengths of our study include the longitudinal study design, a two-year postoperative follow up period, and comparison with a NS control group. Also, we used robust methods to assess body composition (DXA) and mREE (indirect calorimetry). This study has a few limitations. The sample size is relatively small, although, after reverse calculations based on variance observed in our data, our surgical cohort size is enough to see a one-sample difference from the mean of zero with 80% power and 5% alpha error. However, to detect differences between groups (SG and NS) in metabolic adaptations, we will need a sample size of 300, which is practically very challenging given the rates of bariatric surgery utilization. The small sample size also limits our ability to do more-involved analyses and modeling that include other variables that may affect REE (sleep, diet, and stress). Furthermore, a lack of a non-surgical control group that achieved a similar amount of weight loss the surgical cohort would enable dissect if the observed changes in REE were from the SG or from the associated weight loss. Furthermore, we did not have data on all confounding variables, especially dietary changes, which can affect resting energy expenditure. Given the paucity of the literature, conflicting evidence, and the importance of REE in weight control, more research is needed to shed light on the impact of changes in body composition, energy expenditure, and metabolic adaptation following MBS on long-term weight loss.

## 5. Conclusions

We found a decrease in REE and REE normalized for fat-free mass over two years following SG, which may impede post-surgical weight loss (metabolic adaptation was observed at one year and then not at two years). This suggests that there should be emphasis on improving the quantity and quality of fat-free mass to reduce the decreases in REE after SG. Further studies are needed to understand and recognize factors that lead to weight regain in order to advise patients regarding prevention of weight regain after surgery and to devise adjunctive therapies for optimal weight loss.

## Figures and Tables

**Figure 1 nutrients-17-00075-f001:**
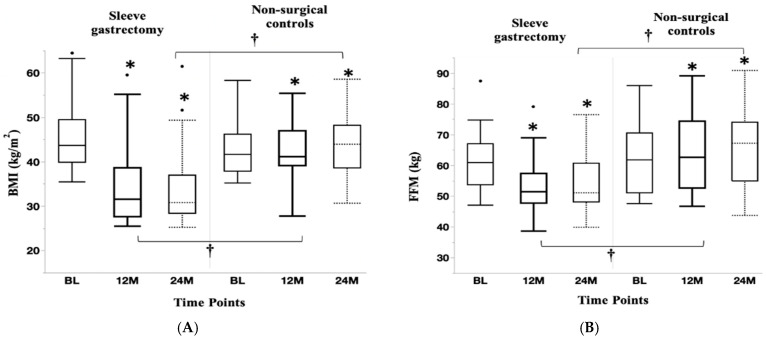
Changes in Body Mass Index (**A**) and fat-free mass (**B**) over two years: BMI and fat-free mass decreased markedly in the first year in the sleeve gastrectomy group, followed by stabilization of these measures between the first and second year. In contrast, BMI and fat-free mass increased over two years in the non-surgical group. * *p* < 0.05 for within-group change from BL; † *p* < 0.05 for between-group change.

**Figure 2 nutrients-17-00075-f002:**
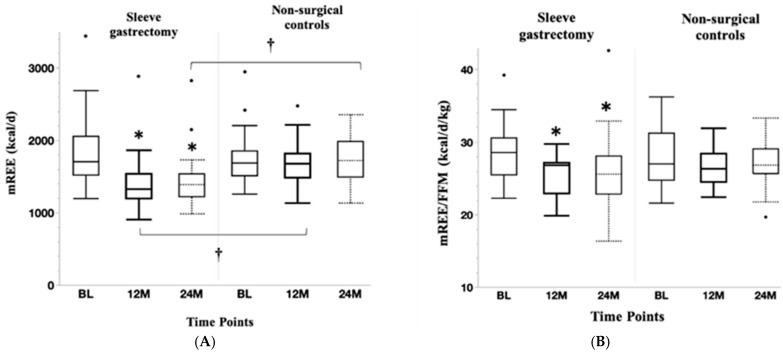
Changes in mREE (**A**) and mREE standardized for FFM (**B**) over two years: mREE and mREE/FFM decreased markedly between baseline and one year followed by stabilization between one and two years. No changes in these measures were noted in the non-surgical controls. Groups differed for mREE, but not mREE/FFM at one and two years. * *p* < 0.05 for within-group change; † *p* < 0.05 for between-group change.

**Table 1 nutrients-17-00075-t001:** Weight Altering Medications at Baseline and Follow Up.

	Sleeve Gastrectomy (SG)		Non-Surgical Controls (NS)	
Medication	Baseline	Follow-Up	Baseline	Follow-Up
ADHD Medications	2		2	
Topiramate	1	1	1	
Phentermine		2		1

**Table 2 nutrients-17-00075-t002:** Baseline demographic and anthropometric characteristics.

		Sleeve Gastrectomy (*n* = 24)	Non-Surgical Controls (*n* = 28)	*p*-Value
Age, years		18.1 ± 0.4	18.0 ± 0.6	0.930
Sex, *n* (%)	Male	4 (20)	7 (25)	0.460
	Female	20 (80)	21 (75)	
Race, *n* (%)	White	14 (58)	14 (50)	0.620
	Black	6 (25)	5 (18)	
	Other/Unknown	4 (17)	9 (32)	
Ethnicity, *n* (%)	Non-Hispanic	15 (62.5)	15 (53.6)	0.440
	Hispanic	9 (37.5)	13 (46.4)	
Height, cm		168.0 ± 2.0	167.4 ± 1.4	0.923
Weight, kg		126.4 (106.5, 145.8)	120.2 (98.2, 131.3)	0.145
BMI, kg/m^2^		43.6 (39.8, 49.4)	41.6 (37.8, 48.2)	0.051
BMI z-score		2.47 (2.30, 2.61)	2.43 (2.29, 2.73)	0.920
BMI expressed as percentage of the 95th percentile of BMI for age and sex		149.0 (139.0, 162.4)	138.8 (131.3, 153.1)	0.053
Sleeping hours, h/wk		55.3 ± 2.6	54.8 ± 1.9	0.890
Vigorous exercise, h/wk		2.0 (0.0, 5.0)	3.8 (0.8, 7.4)	0.210

Data presented as mean ± SE or median (first quartile, third quartile), SG: sleeve gastrectomy group, NS: non-surgical group, BMI: Body Mass Index.

**Table 3 nutrients-17-00075-t003:** Baseline body composition and resting energy expenditure.

	SG (*n* = 24)		NS (*n* = 28)		*p*-Value
% Fat mass	22	50.9 (46.5, 51.9)	27	47.1 (41.1, 48.9)	**0.011**
% Lean mass	22	47.1 (46.2, 51.7)	27	51.4 (49.0, 58.9)	**0.008**
FFM, kg	23	61.8 ± 2.0	27	62.5 ± 2.2	0.805
FFMI, kg/m^2^	23	22.2 ± 0.6	27	22.2 ± 0.5	0.979
Fat mass, kg	22	59.7 (51.5, 71.5)	27	51.4 (45.8, 62.2)	**0.036**
mREE, kcal/d	24	1706.5 (1514.3, 2049.5)	28	1688.0 (1509.0, 1856.0)	0.720
mREE/FFM, kcal/d/kg	23	28.6 ± 0.8	27	28.0 ± 0.7	0.611
mREE/TBW, kcal/d/kg	24	14.2 ± 0.4	28	14.8 ± 0.4	0.356

Data presented as mean ± SE or median (first quartile, third quartile). Bold font denotes a significant *p*-value of <0.05. SG: sleeve gastrectomy group, NS: non-surgical group, FFM: fat-free mass, FFMI: fat-free mass index, mREE: measured resting energy expenditure, TBW: total body weight, RQ: respiratory quotient.

**Table 4 nutrients-17-00075-t004:** Changes in anthropometric measurements and body composition over two years.

Variable	Sleeve Gastrectomy		Non-Surgical Controls		*p*-Value (Within Group)		*p*-Value (Between Group)
	*n*		*n*		SG	NS	
Weight, kg							
One year–BL	22	−31.2 (−45.4, −25.4)	24	3.1 (−1.1, 5.2)	**<0.0001**	0.149	**<0.0001**
Two years–BL	24	−33.0 (−40.1, −24.2)	28	8.3 (1.3, 11.2)	**<0.0001**	**0.003**	**<0.0001**
BMI, kg/m^2^							
One year–BL	22	−11.7 (−15.3, −9.0)	24	0.6 (−0.8, 1.6)	**<0.0001**	0.487	**<0.0001**
Two years–BL	24	−12.4 (−14.4, −9.8)	28	2.2 (0.3, 3.5)	**<0.0001**	0.007	**<0.0001**
BMI z-score							
One year–BL	21	−0.74 (−1.10, 0.53)	24	−0.06 (−0.10, 0.02)	**<0.0001**	0.125	**<0.0001**
Two years–BL	22	−0.85 (−1.05, −0.27)	22	−0.07 (−0.14, 0.01)	**<0.0001**	0.056	**<0.0001**
%BMI of 95th percentile for age and sex, %							
One year–BL	22	−42.5 (−57.3, −34.6)	16	−2.6 (−6.3, 0.9)	**<0.0001**	0.149	**<0.0001**
Two years–BL	24	−43.4 (−53.1, −34.7)	27	0.9 (−3.8, 10.6)	**<0.0001**	0.726	**<0.0001**
Fat mass, kg							
One year–BL	20	−25.4 (−35.3, −16.4)	23	1.4 (−2.7, 3.4)	**<0.0001**	0.496	**<0.0001**
Two years–BL	19	−25.4 (−31.2, 14.4)	25	5.1 (0.3, 7.3)	**0.0005**	0.052	**<0.0001**
FFM, kg							
One year–BL	21	−9.5 (−11.5, −6.1)	23	2.0 (−0.1, 3.0)	**<0.0001**	**0.002**	**<0.0001**
Two years–BL	20	−9.8 (−12.6, −3.7)	25	3.9 (1.5, 5.5)	**0.0012**	**0.007**	**<0.0001**
Percent Total fat mass (%)							
One year–BL	20	−9.9 (−14.3, −4.3)	23	0.4 (−2.8, 1.0)	**<0.0001**	0.297	**<0.0001**
Two years–BL	19	−8.3 (−11.2, −2.7)	25	0.1 (−2.1, 2.0)	**<0.0001**	0.927	**<0.0001**
Percent Total lean mass (%)							
One year–BL	20	9.3 (3.8, 13.2)	23	−0.4 (−1.0, 2.6)	**<0.0001**	0.312	**<0.0001**
Two years–BL	19	7.5 (2.7, 10.3)	25	−0.0 (−1.2, 2.8)	**<0.0001**	0.647	**<0.0001**
Sleeping hours, h/wk							
One year–BL	19	−5.7 ± 2.9	24	−5.1 ± 2.3	0.062	**0.032**	0.870
Two years–BL	18	−6.5 (−13.6, 14.4)	25	−3.0 (−8.3, 2.3)	0.917	0.154	0.712
Vigorous exercise, h/wk							
One year–BL	19	0.0 (−1.5, 3.3)	24	0.0 (−2.9, 2.5)	0.609	0.893	0.493
Two years–BL	18	0.1 (−3.6, 7.0)	25	0.0 (−6.5, 1.9)	0.273	0.447	0.273

Data presented as mean ± SE or median (first quartile, third quartile). Bold denotes significant *p*-value < 0.05. SG: sleeve gastrectomy, NS: non-surgical, One year–BL: the difference between one year and baseline; Two years–BL: the difference between two years and baseline, BMI: Body Mass Index, FFM: fat-free mass.

## Data Availability

The data that support the findings are not publicly available but are available from the corresponding author upon reasonable request.

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
