# Peer review of "Metabolic Adaptation and Its Determinants in Adolescents Two Years After Sleeve Gastrectomy"

_nutrients, 2024, doi:10.3390/nu17010075_

Round 1
Reviewer 1 Report
Comments and Suggestions for Authors
This is an excellent study that addresses a significant gap in the literature by evaluating the effects of gastrectomy in severely obese young individuals through a two-year follow-up study. The introduction is well-written, covering all the key aspects of this topic and supported by current references. The objectives are clearly stated, and the study design effectively addresses these objectives. The results are thoroughly described, and the discussion is appropriately written, allowing for a meaningful comparison with previous studies. Finally, the conclusions are well-supported by the results.
Please find specific details below:
2- Regarding the presentation of the results, the authors presented their results in table (mean plus minus standard error) and the graphs in box plot, allowing a deep analysis of the results.
3- About the main question of the study: In the present study, the authors aimed to determinate the real life changes in the resting energy expenditure (REE) in youth and adolescents before and after gastrectomy surgery, which at the moment have only very limited informations about it.
4- About the conclusions: Finally, the conclusions are well-supported by the results, which clearly demonstrated a reduction of body fat and increasing in fat free mass along the 2 years of the study, while the REE presented a reduction in gastrectomy surgery group in comparisom with the control (non-surgical) group.
Author Response
Reviewer 1:
This is an excellent study that addresses a significant gap in the literature by evaluating the effects of gastrectomy in severely obese young individuals through a two-year follow-up study. The introduction is well-written, covering all the key aspects of this topic and supported by current references. The objectives are clearly stated, and the study design effectively addresses these objectives. The results are thoroughly described, and the discussion is appropriately written, allowing for a meaningful comparison with previous studies. Finally, the conclusions are well-supported by the results.
We thank the reviewer for their comments and positive overall feedback.
Please find specific details below:
1- Regarding the Material and Methods, of note, the authors have used the upmost method for measurement of body composition, the dual-energy X-ray absorptiometry (DXA). Similarly, the resting energy expenditure (REE) was measured by indirect calorimetry, which is the gold standard for measurement of REE. The inclusion and exclusion criteria used by the authors permitted an analysis free of bias.
We thank the reviewers’ appreciation of our strong methodology.
2- Regarding the presentation of the results, the authors presented their results in table (mean plus minus standard error) and the graphs in box plot, allowing a deep analysis of the results.
We agree with the reviewer that since the data are not normally distributed, presentation in box plots is representative. Thank you for appreciating that.
3- About the main question of the study: In the present study, the authors aimed to determinate the real life changes in the resting energy expenditure (REE) in youth and adolescents before and after gastrectomy surgery, which at the moment have only very limited information about it.
We agree that the current data on changes after sleeve gastrectomy (SG) in youth and adolescents is very limited and these data are novel.
4- About the conclusions: Finally, the conclusions are well-supported by the results, which clearly demonstrated a reduction of body fat and increasing in fat free mass along the 2 years of the study, while the REE presented a reduction in gastrectomy surgery group in comparison with the control (non-surgical) group.
We thank the reviewer for agreeing with our conclusions.
Reviewer 2 Report
Comments and Suggestions for Authors
The study aims to investigate changes in resting energy expenditure (REE) and metabolic adaptation in adolescents after sleeve gastrectomy (SG) compared to a non-surgical control group over two years. The rationale for the study is sound, given the limited data on metabolic adaptation in adolescents after SG and the potential implications for long-term weight management. However, the execution of the study and the presentation of the findings are flawed.
Specific concerns and recommendations for revision:
Abstract (Line 22): "Evaluate changes in mREE and metabolic adaptation, and body composition in youth after SG vs. non-surgical (NS) controls over two years." This sentence is grammatically incorrect. Rephrase for clarity. Also, the abstract lacks specific results and conclusions, limiting its impact.
Introduction (Line 41-42): "The high prevalence and increase in severe obesity among children and adolescents is concerning (1, 2) and is known to lead to increased health complications in adulthood." The position of references cited here are missing.
Introduction (Line 51): "However some weight regain is common even after MBS following aÄ´ainment of nadir weight (8) , and 10-20% of patients struggle with considerable weight regain after MBS (9)." The spacing around the reference number (8) is inconsistent. Maintain consistent formatting throughout. Also, define "considerable weight regain." Quantify this term for clarity.
Introduction (Line 64): "There are only a few studies that have evaluated changes in REE after SG, and results are conflicting (15, 16)." While the authors mention conflicting results, they don't elaborate on the nature of these conflicts. Provide a more detailed analysis of the existing literature, highlighting the discrepancies and their potential causes.
Participants and Methods (Line 96): "Exclusion criteria for the study included a) use of antipsychotic medications that cause weight gain if treated for less than six months, or if the dosage was not stable for at least two months..." This is unclear. Rephrase for clarity. Were participants on stable doses of antipsychotics for other indications included?
Participants and Methods (Line 102-108): The information on concomitant medications is poorly presented. Create a clear table outlining the medications used in both groups at baseline and during follow-up.
Participants and Methods (Line 119): "Activity levels and sleeping hours were assessed using the Paffenberger questionnaire (25)." Provide more details about the specific Paffenberger questionnaire used and its validity in this population.
Participants and Methods (Line 132): "As described by Wolfe et al., to assess predicted REE (p REE), we used the baseline data of our cohort in a linear regression model using mREE as the dependent variable and FFM as the independent variable." Provide the specific equation derived from the regression analysis.
Results (Line 157): "Baseline demographic and anthropometric characteristics are shown in Table 1. SG vs. NS participants did not differ in age, sex, race, ethnicity, height, and weight." While statistically insignificant, the differences in BMI and BMI percentile are clinically relevant and should be discussed.
Results (Line 182-192): This section is poorly written and difficult to follow. Clarify the presentation of the results. Provide specific p-values for all comparisons. The description of the changes in BMI and FFM between one and two years is unclear.
Discussion: The discussion is superficial and lacks depth. It fails to adequately address the limitations of the study, such as the small sample size, potential confounding factors, and the lack of detailed dietary information. The implications of the findings for clinical practice are not adequately explored.
References: The reference list is incomplete and contains formatting errors. Ensure that all citations are accurate and formatted according to the journal's guidelines.
Comments on the Quality of English LanguageCan be improved.
Author Response
The study aims to investigate changes in resting energy expenditure (REE) and metabolic adaptation in adolescents after sleeve gastrectomy (SG) compared to a non-surgical control group over two years. The rationale for the study is sound, given the limited data on metabolic adaptation in adolescents after SG and the potential implications for long-term weight management. However, the execution of the study and the presentation of the findings are flawed.
We appreciate reviewer’s agreement about the importance of the study and are thankful of the constructive feedback. We have attempted to address the concerns raised.
Specific concerns and recommendations for revision:
Abstract (Line 22): "Evaluate changes in mREE and metabolic adaptation, and body composition in youth after SG vs. non-surgical (NS) controls over two years." This sentence is grammatically incorrect. Rephrase for clarity. Also, the abstract lacks specific results and conclusions, limiting its impact.
Thanks for picking that up. We have edited the grammatical error and also edited the abstract to present specific results and conclusions (keeping the word limit in mind).
Line 22: We evaluated changes in mREE, metabolic adaptation, and body composition in youth after SG vs. non-surgical (NS) controls over two years.
Also edited the conclusions to state: Despite significant decreases in BMI after SG in youth, no metabolic adaptation was present at two years. This implies that by two years, metabolism has reached a steady state and weight changes after this should be addressed in an unbiased way.
Introduction (Line 41-42): "The high prevalence and increase in severe obesity among children and adolescents is concerning (1, 2) and is known to lead to increased health complications in adulthood." The position of references cited here are missing.
We have moved the cited references to the end of the sentence. We are happy to edit further if needed.
Introduction (Line 51): "However some weight regain is common even after MBS following aÄ´ainment of nadir weight (8) , and 10-20% of patients struggle with considerable weight regain after MBS (9)." The spacing around the reference number (8) is inconsistent. Maintain consistent formatting throughout. Also, define "considerable weight regain." Quantify this term for clarity.
We have edited the spacing. Further we have clarified “considerable weight loss”. This is about 75-90% weight regain in 10-20% of the patients. (line 54)
Introduction (Line 64): "There are only a few studies that have evaluated changes in REE after SG, and results are conflicting (15, 16)." While the authors mention conflicting results, they don't elaborate on the nature of these conflicts. Provide a more detailed analysis of the existing literature, highlighting the discrepancies and their potential causes.
We have now elaborated on the potential reasons for the conflicting data. “There are only a few studies that have evaluated changes in REE after SG, and results are conflicting likely from different methodologies used for evaluating REE, different patient populations (with or without diabetes) and different follow up period”. (Line 68)
Participants and Methods (Line 96): "Exclusion criteria for the study included a) use of antipsychotic medications that cause weight gain if treated for less than six months, or if the dosage was not stable for at least two months..." This is unclear. Rephrase for clarity. Were participants on stable doses of antipsychotics for other indications included?
Thanks for seeking clarification. We have added the clarification stating, “they were on a stable dose and stable weight”. Line 103
Participants and Methods (Line 102-108): The information on concomitant medications is poorly presented. Create a clear table outlining the medications used in both groups at baseline and during follow-up.
We thank the reviewer for the suggestion. We have now added the table below to the section.
|
|
Sleeve Gastrectomy (SG) |
Non-Surgical Controls (NS) |
||
|
Medication |
Baseline |
Follow-Up |
Baseline |
Follow-Up |
|
ADHD Medications |
2 |
|
2 |
|
|
Topiramate |
1 |
1 |
1 |
|
|
Phentermine |
|
2 |
|
1 |
Participants and Methods (Line 119): "Activity levels and sleeping hours were assessed using the Paffenberger questionnaire (25)." Provide more details about the specific Paffenberger questionnaire used and its validity in this population.
We have now added some information about this tool. It states, “Paffenberger questionnaire is a self-reported questionnaire that assesses vigorous, moderate and light physical activity. It has good validity and excellent reliability in measuring physical activity when compared to accelerometers” (Line 130)
Participants and Methods (Line 132): "As described by Wolfe et al., to assess predicted REE (p REE), we used the baseline data of our cohort in a linear regression model using mREE as the dependent variable and FFM as the independent variable." Provide the specific equation derived from the regression analysis.
The specific equation generated was : 171.23 + 24.02 (Total Fat free Mass in Kg). We have now added this equation to the methodology (line 150).
Results (Line 157): "Baseline demographic and anthropometric characteristics are shown in Table 1. SG vs. NS participants did not differ in age, sex, race, ethnicity, height, and weight." While statistically insignificant, the differences in BMI and BMI percentile are clinically relevant and should be discussed.
We agree with the reviewer that while BMI and BMI percentile did not reach statistical significance for differences between groups, the numerical values are different. This is mentioned in Lines 172 under the subject characteristics. This is expected as the surgical participants usually have higher BMI that their non-surgical counterparts.
Results (Line 182-192): This section is poorly written and difficult to follow. Clarify the presentation of the results. Provide specific p-values for all comparisons. The description of the changes in BMI and FFM between one and two years is unclear.
Thanks for seeking clarification. We have edited this section. Specific p-values are presented in the table.
All anthropometric measures and body composition decreases were higher in the SG than NS group from BL to one year and BL to two years. Within- group reductions in BMI and fat free mass following sleeve gastrectomy were most marked one year following surgery, after which these measures stabilized between one and two years. In contrast, the non-surgical group had within group increases in BMI and fat free mass over two years
Discussion: The discussion is superficial and lacks depth. It fails to adequately address the limitations of the study, such as the small sample size, potential confounding factors, and the lack of detailed dietary information.
We have edited the discussion extensively and have also added the clinical implications of these data. As per below:
“We show that REE decreases over two years following SG in adolescents with obesity, and that within the SG group, there is metabolic adaptation at one year which is no longer prevalent two years after surgery. Further, metabolic adaptation is positively associated with two-year change in measured REE (mREE) normalized for changes in fat free mass. However, the two-year change in measured REE (normalized for fat free mass and total body weight) and metabolic adaptation did not differ between surgical and non-surgical groups.
There are very few studies of energy expenditure in adolescents following bariatric surgery with a 2-year follow up. One longitudinal, controlled trial in 11 adolescents with obesity reported that basal metabolic rate declined by 1.5 months and remained suppressed at 6 and 12 months after Roux-en-Y gastric bypass (RYGB). Metabolic adaptation was not investigated in that study. Another longitudinal study with 12 adolescents and young adults used indirect calorimetry to show that REE decreased over the first year following surgery, consistent with a decrease in total lean mass; however, REE standardized for total body weight increased one year after SG . In another study of 20 adolescents undergoing RYGB or SG, mREE decreased by 548 kcal/d one year after surgery. We found similar decreases in measured REE in our study (decreased by 401 kcal/d two years) after SG. Further we found that there was metabolic adaptation (measured REE was lower than predicted REE based on body composition) at one-year but this was no longer prevalent at two years. Metabolic adaptation was present during the first year after surgery during the active weight loss stage and then stabilized at two years when weight loss plateaued despite a lower weight state as compared to presurgical weight.
Significant reductions in REE normalized for fat free mass and metabolic adaptation after MBS could be one of the contributing factors to the variable weight loss seen after sleeve gastrectomy and may contribute to weight regain. According to one study, it is possible that changes in muscle mass that occur after MBS could play a crucial role in alterations in REE as well as the degree of weight loss and weight loss maintenance. Data suggests that irrespective of the modality of weight loss – very low-calorie diet or bariatric surgery, the degree of metabolic adaptation is associated with degree of increase in appetite. This is significant clinically as changes in appetite after MBS can be monitored and may be predictive of weight loss.
Weight loss during the first year after surgery includes reductions in fat mass and fate free mass, with greater reductions occurring in fat mass as compared to fat free mass. Most of the reductions in the metabolically active fat free mass occur from reductions in skeletal muscle. Some studies have found that the REE/FFM ratio decreases after MBS, while the REE/TBW ratio increases, postulating that the decrease after surgery in REE/FFM could be due to a decline in FFM metabolism, while the increase in REE/TBW following surgery is primarily driven by the substantial loss of FM. We report similar results in our study with decreases in mREE/FFM but no changes in mREE/TBW at one and two years after SG.
Data on metabolic adaptation after MBS in adults are conflicting, with studies reporting decreases, increases, or no changes in metabolic adaptation post-surgery. One study reported that metabolic adaptation at six months was different from baseline with no further changes between six months and two years. Consistent with the literature, in our study, we report metabolic adaptation at one year, but not at two years after sleeve gastrectomy. The inconsistency in results across studies likely originates from the duration of follow-up, with shorter follow-up studies reporting metabolic adaptation when there is a physiological effort to reduce/prevent weight loss and this is not seen in longer-term (one year and longer) studies. Differences in methodologies to estimate REE and metabolic adaptation, small sample size and differences in the weight loss modality are other possible explanations for conflicting data.
In our study, we show that the degree of metabolic adaptation is associated with changes in REE normalized to fat free mass. Hence decreasing the loss of fat free mass and making the skeletal muscle more metabolically active after surgery may help decrease metabolic adaptation and improve weight loss after surgery. Engagement in physical exercise is associated with greater weight loss after MBS . Activity levels and sleeping hours could be factors influencing postoperative changes in REE. However, in our study neither vigorous exercise nor sleeping hours differed between groups at baseline or follow-up. High protein diet is recommended after surgery and considered to be helpful in preserving lean mass. However, recent study did not show preservation of lean mass after protein supplementation in post-bariatric patients.”
We have appended the limitations of the study including the lack of detailed dietary information.
“The small sample size also limits our ability to do more involved analyses and modeling including other variables that may affect REE (sleep, diet and stress). Further, a lack of a non-surgical control group that achieved a similar amount of weight loss the surgical cohort would enable dissect if the observed changes in REE were from the SG or from the associated weight loss. Further, we did not have data on all confounding variables, especially dietary changes that can affect resting energy expenditure.”
Reviewer 3 Report
Comments and Suggestions for Authors
The article addresses a designed experiment addressing weight loss through bariatric surgery, to evaluate changes in resting energy expenditure and body composition over two years, and differential metabolic adaptation at two years between surgical and non-surgical conditions. The study also examines how changes in resting energy expenditure and metabolic adaptation are associated with changes in weight and body composition. The manuscript updates reported findings on one-year changes.
Research methods employed are appropriate within parametric and nonparametric contexts, although they seem rather basic as reported findings are correlational rather than model-based and do not seem to incorporate information fronm additional variables. Regression results are mentioned, but no detail is provided about modeling procedure or regarding the validity of the predicted values that are used in formulating comparisons.
The references are relevant, although only four appear to have been published in 2020 or later. Updated references would add to the impact of the research and would provide a springboard to further research.
The manuscript generally is well-constructed and readable. However, the conclusions are very sketchy and should incorporate additional information. Specifically, the conclusion hints only very briefly at implications of the findings. The broader impact of the research will be enhanced greatly by adding, say, a couple of paragraphs discussing implications for healthcare policy and for clinical practice.
With multiple hypothesis tests, it would be ideal to account for the resulting inflated Type error level by employing False Discovery Rate adjustments, or possibly the more conservative Bonferroni adjustment.
Author Response
The article addresses a designed experiment addressing weight loss through bariatric surgery, to evaluate changes in resting energy expenditure and body composition over two years, and differential metabolic adaptation at two years between surgical and non-surgical conditions. The study also examines how changes in resting energy expenditure and metabolic adaptation are associated with changes in weight and body composition. The manuscript updates reported findings on one-year changes.
We thank the reviewer for summarizing the study and appreciate their effort in providing the review.
Research methods employed are appropriate within parametric and nonparametric contexts, although they seem rather basic as reported findings are correlational rather than model-based and do not seem to incorporate information from additional variables. Regression results are mentioned, but no detail is provided about modeling procedure or regarding the validity of the predicted values that are used in formulating comparisons.
We thank the reviewer for seeking clarification of statistical methodology. We agree that our analysis is basic and the findings are correlational. We are limited by small sample size which limits the variables that can be added in the regression models. Furthermore, the measures of body composition are highly correlated and adding them int a model will make our model unstable. Hence, we are limited by the exploratory nature of the study and small sample size.
We have elaborated the dependent and independent variables used to generate a regression equation in calculating the predicted REE (lines 146-147)
The references are relevant, although only four appear to have been published in 2020 or later. Updated references would add to the impact of the research and would provide a springboard to further research.
We thank the reviewer for noting this. We reviewed the literature for most recent articles since 2020 that evaluate REE after bariatric surgery and have added the references below. None of studies were done in adolescents and youth. We have added this to the discussion which makes it more substantive. (References: 1. Adaptive thermogenesis, at the level of resting energy expenditure, after diet alone or diet plus bariatric surgery by Torres et al: and 2.The effect of additional protein on lean body mass preservation in post-bariatric surgery patients: a systematic review)
The manuscript generally is well-constructed and readable. However, the conclusions are very sketchy and should incorporate additional information. Specifically, the conclusion hints only very briefly at implications of the findings. The broader impact of the research will be enhanced greatly by adding, say, a couple of paragraphs discussing implications for healthcare policy and for clinical practice.
We thank the reviewer for this suggestion. We have reworded the discussion substantially and added more depth to the interpretation, updated the references, provided clinical relevance of the findings and elaborated on the study limitations.
With multiple hypothesis tests, it would be ideal to account for the resulting inflated Type error level by employing False Discovery Rate adjustments, or possibly the more conservative Bonferroni adjustment.
We agree with the reviewer completely that the validity of the results will improve if we use the false discovery rate correction or the Bonferroni correction. However, we have not done this due to the exploratory nature of the study and limited sample size limiting our power overall.
Round 2
Reviewer 2 Report
Comments and Suggestions for Authors
The manuscript has been well organized and can be accepted.
Reviewer 3 Report
Comments and Suggestions for Authors
The authors seem to have done well in responding to reviewer comments.